# EjFAD8 Enhances the Low-Temperature Tolerance of Loquat by Desaturation of Sulfoquinovosyl Diacylglycerol (SQDG)

**DOI:** 10.3390/ijms24086946

**Published:** 2023-04-08

**Authors:** Xun Xu, Hao Yang, Xiaodong Suo, Mingxiu Liu, Danlong Jing, Yin Zhang, Jiangbo Dang, Di Wu, Qiao He, Yan Xia, Shuming Wang, Guolu Liang, Qigao Guo

**Affiliations:** 1Key Laboratory of Horticulture Science for Southern Mountains Regions of Ministry of Education, College of Horticulture and Landscape Architecture, Southwest University, Chongqing 400715, China; 2Academy of Agricultural Sciences of Southwest University, State Cultivation Base of Crop Stress Biology for Southern Mountainous Land of Southwest University, Chongqing 400715, China

**Keywords:** loquat, triploid, EjFAD8, low temperatures, fatty acid

## Abstract

Loquat (*Eriobotrya japonica* Lindl.) is an evergreen fruit tree of Chinese origin, and its autumn–winter flowering and fruiting growth habit means that its fruit development is susceptible to low-temperature stress. In a previous study, the triploid loquat (B431 × GZ23) has been identified with high photosynthetic efficiency and strong resistance under low-temperature stress. Analysis of transcriptomic and lipidomic data revealed that the fatty acid desaturase gene *EjFAD8* was closely associated with low temperatures. Phenotypic observations and measurements of physiological indicators in *Arabidopsis* showed that overexpressing-*EjFAD8* transgenic plants were significantly more tolerant to low temperatures compared to the wild-type. Heterologous overexpression of *EjFAD8* enhanced some lipid metabolism genes in *Arabidopsis*, and the unsaturation of lipids was increased, especially for SQDG (16:0/18:1; 16:0/18:3), thereby improving the cold tolerance of transgenic lines. The expression of ICE-CBF-COR genes were further analyzed so that the relationship between fatty acid desaturase and the ICE-CBF-COR pathway can be clarified. These results revealed the important role of EjFAD8 under low-temperature stress in triploid loquat, the increase expression of *FAD8* in loquat under low temperatures lead to desaturation of fatty acids. On the one hand, overexpression of *EjFAD8* in *Arabidopsis* increased the expression of ICE-CBF-COR genes in response to low temperatures. On the other hand, upregulation of *EjFAD8* at low temperatures increased fatty acid desaturation of SQDG to maintain the stability of photosynthesis under low temperatures. This study not only indicates that the *EjFAD8* gene plays an important role in loquat under low temperatures, but also provides a theoretical basis for future molecular breeding of loquat for cold resistance.

## 1. Introduction

The loquat is an evergreen fruit tree that originated in south-western China and has had a long history of cultivation in China for over 2000 years [1]. Loquat is native to the subtropical regions of humid heat, and its phylogeny is influenced by climatic environments, making it very sensitive to low temperatures [2,3]. The loquat differs from other fruit trees in that its phenological period is generally autumn–winter flowering and fruiting, so its flower and juvenile fruit development is highly susceptible to low-temperature stress, resulting in reduced yields [4,5]. With the recent increase in extreme weather, low temperatures have become one of the most common stressors in plants, with cold and frost damage greatly impacting plant growth and development [6]. Thus, it is urgent to study the molecular regulatory mechanisms of loquat under low temperatures.

When plants are exposed to low temperatures, the cell membrane is the first to sense the low temperatures. Therefore, the cell membrane play a very important role in the resistance of plants to low temperatures [7]. Another role of cell membranes is as an important barrier to reduce cold stress damage by increasing the mobility of membrane lipids [8,9]. The fluidity of membrane lipids is mainly determined by the level of unsaturated fatty acids and phospholipids. Levels of unsaturated fatty acids and phospholipids increase under low temperatures, effectively reducing cold-induced electrolyte leakage and cell dehydration [10,11]. Fatty acids are important components of membrane lipids, including saturated fatty acids, monounsaturated fatty acids, and polyunsaturated fatty acids [12]. Fatty acid desaturases (FADs) are key metabolic enzymes in lipid metabolism that introduce double bonds into fatty acids at specific sites to transform saturated fatty acids into unsaturated fatty acids [13]. The expression levels of the genes encoding fatty acid desaturases represent their content. The ω-3 fatty acid desaturases, representative of the fatty acid desaturase family, transform diene fatty acids (DA) into triene fatty acids (TA), in which FAD3 is located in the endoplasmic reticulum (ER) and FAD7 and FAD8 in the plastid [14,15,16].

FAD8 is a very important enzyme among the ω-3 fatty acid desaturases [17]. In chloroplasts, FAD7 and FAD8 desaturated the 18:2 fatty acid acyl chains in MGDG, DGDG, SQDG, and PG into 18:3 fatty acid acyl chains [18].The transcript levels of *FAD8* are influenced by the environment. *FAD8* has been found and identified in many plants to be upregulated at low temperatures and down-regulated at high temperatures. For example, *AtFAD8* was up-regulated in *Arabidopsis* at 8 °C and down-regulated at 30 °C; rice lacking *OsFAD8* was more susceptible to damage at low temperatures, suggesting that *FAD8* plays an important role after plants are subjected to low-temperature stress [18,19]. In contrast, *ZmFAD8* was up-regulated at 50 °C, reflecting the thermal sensitivity of *ZmFAD8* [20].

Low temperatures greatly affect the metabolism and transcriptome of plants, and plants have evolved a complex response mechanism in order to withstand low temperatures. Low-temperature tolerance is regulated by a wide range of genes in plants, and the low temperatures transcription regulatory network composed of ICE-CBF-COR is the most widely studied pathway of plant responses to low temperatures [21,22]. C-REPEAT/DRE BINDING FACTORs (*CBFs*) are important transcription factors in plant response to low temperatures and play a very central role in the low-temperature stress regulatory network of plants [23,24]. *ICE* induces transcriptional activation of *CBF* gene family expression at low temperatures. The induced CBF protein binds to the cis-element present in the COR genes promoter and activates the expression of the COR genes, thus increasing the cold tolerance of the plant [25]. Chloroplasts also participate in the ICE-CBF-COR regulatory pathway. Previous studies have shown that the expression of *COR* genes was regulated by the redox status in chloroplast [26,27]. Fatty acid desaturation plays an important role in chloroplasts as an important redox reaction. Li et al. showed that overexpressing ScCBF1 could increase the expression of fatty acid desaturase genes; this result could also indicate that fatty acid desaturation genes were associated with the ICE-CBF-COR regulatory pathway [28].

In our previous studies, the F1 generation of triploid loquats obtained by crossing the tetraploid loquat (B431) with the diploid wild loquat (GZ23) was more low-temperature-tolerant than its parents [29]. We then conducted a series of studies on its physiology and biochemistry. In this study, we obtained the differentially expressed gene *EjFAD8* by transcriptome and lipidome analysis of loquat leaves before and after low-temperature stress. Studies on EjFAD8 in loquat have not been reported. Subsequently, we cloned *EjFAD8* and transferred it into *Arabidopsis* to observe its effect on tolerance to low-temperature stress by phenotypic observation, expression assay, and physiological indicators. These results revealed that EjFAD8 enhances low-temperature tolerance in *Arabidopsis.* It improved the stability of chloroplasts at low temperatures by increasing the unsaturation of SQDG fatty acids in chloroplasts, thus maintaining photosynthesis in plants under low temperatures. In addition, at certain degrees our study provides new insights into the molecular regulatory mechanisms of loquat under low-temperature stress.

## 2. Results

### 2.1. EjFAD8 Plays an Important Role in Lipid Metabolism in the Triploid Loquat at Low Temperatures

To explain the greater cold resistance of triploid loquats compared to their parents, we analyzed differentially expressed genes (DEGs) related to lipid metabolism in these loquats after exposure to low-temperature stress. In the pathway of fatty acid desaturation and synthesis, the two genes encoding fatty acid desaturases, *EjFAD7* and *EjFAD8*, were significantly up-regulated in both the triploid and its parental loquat (Figure 1(Ad)). Among them, *EjFAD8* was more significantly up-regulated in the triploid loquat than in its parental loquat, while there were no significant differences in the other three pathways (Figure 1(Aa–c)).

In additional, for verifying the changes of specific genes in the lipid metabolic network, a schematic representation of the gene network was plotted that associated the lipid metabolism in these three different ploidy loquats under low-temperature stress. (Figure 1B). In the chloroplast outer envelope, the (16:0/18:2) and (18:2/18:2) fatty acid acyl chains in MGDG and DGDG were generated by the combination of EjFAD7 and EjFAD8 to produce the (16:0/18:3) and (18:3/18:3) fatty acid acyl chains. In the chloroplast inner envelope, SQDG (16:0/18:2), SQDG (18:2/18:2), and PG (16:1/18:2) were also generated by EjFAD7 and EjFAD8 acting together to synthesize SQDG (16:0/18:3), SQDG (18:3/18:3), and PG (16:1/18:3). As shown in Figure 2, all lipids showed significant up-regulation in triploid loquat at low temperatures, except for DGDG (18:3/18:3). In particular, the quantity of DGDG (16:0/18:3), MGDG (18:3/18:3), SQDG (16:0/18:3), SQDG (18:3/18:3), and PG (16:1/18:3) in the triploid loquat was significantly up-regulated under low-temperature conditions. These lipid results were consistent with the transcriptomic data that EjFAD8 was most significantly upregulated in triploid loquat after exposure to low temperatures.

The results showed that the response of EjFAD8 to low temperatures caused an increase in unsaturated fatty acids such as MGDG, DGDG, PG, and SQDG in triploid loquat compared to diploid and tetraploid loquat.

### 2.2. Characterization of EjFAD8 from Loquat

*EjFAD8* was annotated as a fatty acid desaturases gene in the loquat genome, and 1155 bp fragment was amplified from the cDNA of triploid loquat leaves, named *EjFAD8*. Sequence comparison of the FAD8 protein in nine plants showed that the FAD8 protein is conserved and has three very conserved transmembrane domains (Figure 3A). Analysis of phylogenetic tree construction branches revealed that EjFAD8 has the highest homology with PbFAD8 in white pear (*Pyrus bretschneideri*) and MdFAD8 in apple (*Malus domestica*) in the Rosaceae family, which could also indicate that FAD8 is a highly evolutionarily conserved protein in higher plants (Figure 3B).

Subsequently, the expression of *EjFAD8* was analyzed by qPCR in loquats of different ploidy at low temperatures (Figure 3C). The diploid, triploid, and tetraploid loquats were all found to show a distinct increase in expression at 0 °C, but the most significant up-regulation was in the triploid, followed by the tetraploid. It can be concluded that EjFAD8 may be positively correlated with low-temperature resistance in loquats, especially in triploid loquats.

### 2.3. Subcellular Localization of EjFAD8

To investigate the subcellular localization of EjFAD8, the recombinant fusion vector and the control empty vector were transferred into young tobacco leaves by *Agrobacterium*-mediated transformation. As shown in Figure 4, the green fluorescence of the control was mainly localized in the cell membrane and nucleus, while the signal of 35S::*EjFAD8-EGFP* was only present in the chloroplast. The findings indicated that EjFAD8 is a chloroplast-localized protein.

### 2.4. Overexpression of EjFAD8 Enhances Low-Temperature Tolerance in Arabidopsis

As the genetic transformation system of loquat has not been established, *EjFAD8* was transferred to *Arabidopsis* in order to further analyze the function of *EjFAD8*. Plant expression vectors for *EjFAD8* were constructed, and genetically transformed plants were obtained by *Agrobacterium*-mediated transformation. Three overexpression lines of *EjFAD8* were obtained by molecular identification and expression analysis (Appendix A).

OE6, OE7, and WT were incubated at 23 °C and 4 °C for 10 days. It could be observed that the root lengths of WT and transgenic lines were between 7 cm and 10 cm at 23 °C (Figure 5A,E), while the OE6 lines had a slightly shorter root length. However, after 10 days of incubation at 4 °C, the average root length of WT was only 0.98 cm, and the average root length of OE6 and OE7 was 1.62 cm and 1.73 cm (Figure 5B,F). The results concluded that the root growth of the transgenic lines was significantly longer than that of WT at 4 °C. It could be seen that the growth potential of the transgenic lines at 4 °C was higher than that of WT.

As shown in Figure 5D, significant wilting and death of WT leaves occurred in *Arabidopsis* treated at 0 °C. OE6 and OE7 suffered from leaf dehydration, but the leaves still grew well after recovering to normal temperature. In previous studies, the content of H_2_O_2_, chlorophyll content, and ion leakage could be used as important parameters of plant low-temperatures tolerance, so nitroblue tetrazolium (NBT) and 3,3′-diaminobenzidine (DAB) were used to stain *Arabidopsis* leaves to observe ROS levels in transgenic lines versus WT. As shown in Figure 5C, there was no significant ROS accumulation in either WT or transgenic lines in the control group. When exposed to low temperature conditions at 0 °C, significant levels of O^2−^ and H_2_O_2_ could be detected in WT plants, while ROS levels could only be partially examined in OE6 and OE7. To assess the membrane integrity of *Arabidopsis* leaves at low temperatures, the electrolyte permeability of these leaves was subsequently measured. At normal temperature, the WT and transgenic lines had the same electrolyte permeability, and the transgenic lines had significantly lower electrolyte permeability than the WT under 0 °C (Figure 5G). The chlorophyll content of the leaves of these *Arabidopsis* lines was then measured and found to be higher in OE6 and OE7 than in WT at 0 °C (Figure 5H). In summary, overexpression of *EjFAD8* enhanced low-temperatures tolerance in *Arabidopsis* leaves.

### 2.5. Overexpression of EjFAD8 under Low-Temperature Stress Increases Transcription of Low-Temperature-Tolerant Genes in Transgenic Arabidopsis

To assess the effect of *EjFAD8* on low-temperature tolerance in transgenic lines, seven genes associated with low-temperature stress were selected in *Arabidopsis*, and their relative expression was measured at 23 °C and 0 °C for 3 days (Figure 6). Three *Arabidopsis* C-REPEAT/DRE BINDING FACTORs (*AtCBF1*, *AtCBF2*, and *AtCBF3*) were up-regulated in transgenic lines and WT after exposure to low temperatures, and the up-regulation was significantly higher in transgenic lines than in WT. *AtCBF3* expression was significantly lower in transgenic lines than in WT at 23 °C, but at low temperatures the results were reversed. *AtICE1* transcript levels were lower in OE6 and OE7 than in WT at 23 °C, but higher in transgenic lines after exposure to low temperatures. The expression of the three target genes of *AtCBFs*, *AtCOR47*, *AtCOR15A*, and *AtRD29A* were also measured. It was observed that all three genes had a significant up-regulation in the transgenic lines relative to WT after being subjected to low temperatures.

These results showed that overexpression of *EjFAD8* could significantly change the expression of low-temperature-tolerant genes. This indicates that *EjFAD8* increased the low-temperature tolerance of plants by increasing the expression of these low-temperature responsive genes.

### 2.6. FAD8-Related Lipids Are Increased in Transgenic Arabidopsis under Low-Temperature Stress

Based on our above analysis of lipidomic data from loquat at different ploidy levels, further analysis of lipid changes was subsequently carried out in transgenic lines and WT at low temperatures. A number of lipids associated with FAD8 metabolism were analyzed, as shown in Figure 7A. Under 0 °C, the quantity of DGDG (16:3/18:3), DGDG (18:3/18:3), SQDG (16:0/18:1), and SQDG (18:3/18:3) were reduced in WT, whereas the quantity of these lipids was increased in the transgenic lines. SQDG (16:0/18:3) increased in both wild-type and transgenic lines at low temperature, but the most significant increase was observed in OE6. However, MGDG (16:0/18:3) was down-regulated in both wild-type and transgenic *Arabidopsis* after low-temperature treatment. The fold change of MGDG (18:2/18:2) and MGDG (18:3/18:3) at low temperature was also not significantly different in wild-type and transgenic *Arabidopsis*. In order to analyze the relationship between these lipid metabolisms and gene expression, expression analyses of the genes involved in these lipid metabolisms were subsequently performed (Figure 7B). The expression of *AtFAD8* in WT treated at 0 °C for 3 days was only 1.2 times higher than at 23 °C, whereas the expression of *AtFAD7* was only half that of the control. The response of *EjFAD8* to low temperatures and the significant up-regulation of *AtFAD8* in the overexpression lines led to an increase in the quantity of their associated metabolic lipids. Furthermore, the presence of unsaturated lipids in SQDG was significantly up-regulated in transgenic lines at low temperatures, from which it could also be assumed that EjFAD8 is SQDG-preferring.

To understand the effect of EjFAD8 on other lipids in the *Arabidopsis* lipid metabolism pathway, expression analysis of *AtFAD4*, *AtFAD6*, *AtMGDs*, *AtSQDs*, *AtDGDs*, and *AtPGP1*, which are involved in these pathways, was performed (Figure 7B). Both *AtFAD4* and *AtFAD6* were involved in the desaturation of the fatty acid chain from 18:0 to 18:2. *AtFAD4* and *AtFAD6* were both down-regulated in the transgenic lines at 23 °C, but OE6 was significantly up-regulated in the transgenic lines after treatment at 0 °C. *AtMGDs* are jointly involved in the pathway from DAG to MGDG, and it can be seen that *AtMGD1* is significantly down-regulated in the transgenic lines compared to WT after exposure to low temperature, while *AtMGD2* and *AtMGD3* are higher relative to WT. *AtPGP1*, *AtDGDs*, and *AtSQDs* are all involved in the DAG to PG, DGDG, and SQDG production pathways, and *AtPGP1*, *AtDGDs*, and *AtSQDs* are all significantly higher in low-temperature transgenic lines than in WT.

The increase in lipids associated with FAD8 metabolism was higher in the transgenic lines than in WT under low-temperature treatment, which is consistent with the above data on lipids in loquat. This result suggests that overexpression of *EjFAD8* enhances the desaturation of fatty acid chains by enhancing the transcription of lipid-related genes in *Arabidopsis*.

## 3. Discussion

Loquat has the biological characteristic of flowering and fruiting in autumn and winter [4]. Its fruit development urgently requires the availability of photosynthetic products. Maintaining the photosynthesis of loquat at low temperatures is the key to ensuring its yield. Therefore, obtaining loquat cultivars with low-temperatures tolerance and high photosynthetic efficiency is the core of our study. Previously, our laboratory obtained the F1 generation of triploid loquat by crossing the tetraploid loquat (B431) with the diploid Guizhou wild loquat (GZ23), which, after a series of experimental studies, was found to be more low-temperature-tolerant and photosynthetically efficient than its parents [29]. In this study, we used a combination of transcriptomic and lipidomic approaches to select from them a key gene, *EjFAD8*, in response to low temperatures in triploid loquats (Figure 1 and Figure 2). Based on a study of transcriptomic data and previous literature [30,31], schematic diagrams were drawn to reveal the lipid gene metabolic network in different ploidy loquat leaves under low-temperature stress (Figure 1B). Analysis of the lipid data also confirmed that lipids derived from EjFAD8 anabolism were significantly higher (MGDG, DGDG, SQDG, and PG) in triploid loquats (Figure 2). Analysis of the expression of different ploidy loquats after treatment at low temperatures showed that *EjFAD8* was up-regulated in loquats at low temperatures (Figure 3C). This result, combined with the above transcriptomic data as well as the lipidomic data, leads us to infer that EjFAD8 plays an important role in improving low-temperature tolerance in the triploid loquat.

When plants were exposed to low-temperature stress, cold signals were transmitted from the cell membrane into the nucleus and the cells underwent a series of physiological and biochemical changes, which induced the expression of many genes associated with the low-temperature response [21]. The expression of these genes was transmitted to the chloroplast through molecular signals and triggered changes in the expression of low-temperature responsive genes in the chloroplast. Plants increase their low-temperature tolerance due to changes in gene expression [32]. The FAD8 protein has an important role in plant lipid metabolism. It is implicated in the synthesis of triene fatty acids (TA) in chloroplasts and acts mainly on the 18:2 fatty acid acyl chains to desaturate them to 18:3 fatty acid acyl chains [18]. By subcellular localization it can be observed that, as in other plants, EjFAD8 is localized in the chloroplast (Figure 4). Low temperatures disrupt the photosynthetic system of plants and affect the development of chlorophyll. In this study, we found that the content of chlorophyll was higher in transgenic plants than in WT under low temperatures (Figure 5H). Previous studies have shown that triploid loquats can also show high photoenergy utilization efficiency and regulation ability at low temperatures [29]. After exposure to low temperatures in plants, the cell membrane received cold signals that triggered changes in intracellular molecular regulatory mechanisms. Upon sensing cold signals in chloroplasts, many low-temperature responsive genes, including *FAD8*, were induced. *FAD8* is a fatty acid desaturase gene localized within the chloroplast, closely related to the unsaturation of the fatty acids MGDG, SQDG, DGDG, and PG. Our study also showed that heterologous overexpression of the *EjFAD8* gene caused a significant increase in the unsaturation of SQDG under low-temperature stress. In addition, lipids such as SQDG are important lipids on chloroplast membranes, and chlorophyll content was significantly up-regulated in overexpressing *Arabidopsis* [33]. It was hypothesized that heterologous overexpression of *EjFAD8* in *Arabidopsis* might lead to better cold resistance in *Arabidopsis* by increasing the unsaturation of lipids such as SQDG and chloroplast content. Thus, such enhanced protection of the chloroplast structure promotes photosynthesis and increases plant resistance to low temperatures [34]. It was speculated that EjFAD8 maintained the photosynthesis of the plant at low temperatures, and this may be one of the reasons why the photosynthetic efficiency of the triploid loquat was higher than its parent loquat.

The FAD8 protein is highly specific for 18:2 fatty acid chains in chloroplasts [18]. In addition, the FAD8 protein responds to low temperatures and thus functions better as a fatty acid desaturase at low temperatures. To verify the role of EjFAD8 in *Arabidopsis*, we then analyzed lipid data in *Arabidopsis* and found that the increase in 18:3 fatty acids was higher in overexpressing *Arabidopsis* at low temperatures than in WT (Figure 7A), and it can be seen that the unsaturated fatty acid content is significantly higher in the SQDG. Based on past research that illustrated the preference of FAD8 for PG [18], it was hypothesized that FAD8 also has a preference for SQDG. Subsequently, we quantified the genes involved in unsaturated fatty acid synthesis in *Arabidopsis* by qPCR and observed that the expression of genes associated with unsaturated fatty acid synthesis was almost always higher in transgenic plants than in the wild type after exposure to low temperature (Figure 7B). This study provided a deeper understanding of the role played by the EjFAD8 protein in lipid metabolism.

Analysis of lipid data in *Arabidopsis* revealed a significant increase of unsaturated SQDG (16:0/18:3) in transgenic plants under low temperature. This result was consistent with that of loquat lipid. It could be speculated that *EjFAD8* was the main reason for the increase of unsaturated SQDG in loquat under low temperature. Studies of other species have shown that SQDG was important for resistance and maintenance of photosynthesis in plant. Loss of SQDG production decreased grana stacking in the leaf chloroplasts of *Arabidopsis* [35], and SQDG shows a very significant influence on the activity of PSII acceptors and donors, which affects photosynthesis [36]. SQDG can be desaturated by FAD8; this redox reaction could affect the expression of *COR* genes in the ICE-CBF-COR pathway in response to low temperatures [37]. ICE-CBF-COR is the most important mechanism for plant response to low temperatures [38]. *COR*, *KIN1*, and *RD29A* are target genes of *CBFs*, while *ICE1* is a positive regulator of *CBFs* [39]. In this study, the expression of ICE-CBF-COR genes in *Arabidopsis* was determined, and it could be shown that the expression of these key genes was significantly higher in transgenic plants than in WT at low temperatures (Figure 6). This suggests that the EjFAD8 protein enhances the expression of low temperature resistance genes and is thereby involved in ICE-CBF-COR, a low temperatures tolerance mechanism.

Chloroplast could produce large amounts of reactive oxygen species under low-temperature stress, which once accumulated in excess can lead to oxidative stress and cell death [40,41]. In the study, we found that transgenic lines had better tolerance to low temperatures compared to WT. The staining of leaves for ROS with DAB and NBT also showed that the leaves of transgenic plants accumulated significantly less reactive oxygen species than WT (Figure 5C). Lee found that overexpression *FAD8* in sweet potato (*Ipomoea batatas* [L.] Lam) reduced the accumulation of reactive oxygen species [42], which was consistent with our findings and suggested that FAD8 protein may be involved in maintaining the homeostasis of ROS in cells. Ion leakage is an important parameter for evaluating cell membrane damage in plants [43]. Our measurements of ion leakage in leaves revealed less damage to cell membranes in transgenic lines at low temperatures (Figure 5G). It also suggested that EjFAD8 maintained the stability of photosynthesis in loquat through participation in the regulation of ROS in chloroplasts, and this may be one of the reasons why the photosynthetic efficiency of the triploid loquat was higher than its parent loquat.

In conclusion, our results showed that *EjFAD8* is a very useful gene that plays a very important role in the resistance of loquat to low-temperature attack. It enhanced the low-temperature tolerance of plants by increasing the 18:3 unsaturated fatty acid content in chloroplast SQDG and the expression of low-temperature responsive genes. Meanwhile, EjFAD8 maintained the stability of photosynthesis and better stress resistance in triploid loquat by affecting the pathway of ICE-CBF-COR and the regulation of ROS. It also provided new insights for molecular markers and molecular breeding in loquat.

## 4. Materials and Methods

### 4.1. Plant Materials and Growth Conditions

The triploid loquat is the F1 generation of B431 × GZ23, the tetraploid loquat is the parent of Longquan 1 (B431), and the diploid loquat is the parent of Guizhou wild loquat (GZ23). All these loquat materials were grown in the Loquat Resource Garden at Southwest University, Chongqing, China. The trees involved in the experiment were all grafted seedlings using the loquat cultivated species as rootstock and the three ploidy loquats mentioned above as scions. Each ploidy of loquat was treated at −3 °C for 72 h at a controlled temperature of 25 °C, and other culture conditions were kept constant. Four biological replicates were carried out.

These plant materials were also used in this experiment (*Arabidopsis thaliana*, wild-type (WT) Col-0, and overexpressing-*EjFAD8* transgenic lines). After vernalizing at 4 °C for 3 days, seeds were first sown on a substrate (3:1:1 mixture of grass charcoal, vermiculite, and perlite) or sterile 1/2 Murashige and Skoog (1/2 MS) medium, and subsequently placed in an artificial climate chamber (23 °C, 16 h light, 70% humidity). The seeds were rigorously sterilized before sowing on 1/2 MS medium. Screen transgenic seeds were processed as follows: sterilized with 75% ethanol for 2 min, washed with 1‰ Tween 80 for 8 min, then sterilized four times with 75% ethanol and washed once with sterile water and spread on 1/2 MS medium with 50 mg/L of kanamycin antibiotic.

### 4.2. Cloning and Sequence Analysis of EjFAD8

After selecting *EjFAD8* (EVM0014726) from the transcriptome data, they were compared with the sequencing results of Jing et al. [44]. The coding sequence (CDS) of *EjFAD8* was found in the loquat coding sequence database, and a pair of gene-specific primers was designed based on its sequence [45]. The full-length cDNA of *EjFAD8* was obtained by PCR amplification, and the PCR amplification product was then ligated to the pTOPO-Blunt simple Cloning Vector and transferred to the *E. coli* strain for DNA sequencing. The specific primers are listed in Appendix A.

Multiple sequence comparisons were performed using Clustal X calibrated sequences and displayed by Jalview software; phylogenetic analysis was constructed by MEGA software under the neighbor-joining method with 1000 bootstrap repeats [46]; other sequences were obtained from NCBI (https://www.ncbi.nlm.nih.gov/ (accessed on 11 July 2022)), and the gene IDs are listed in Appendix A.

### 4.3. Arabidopsis Transformation

The protein sequence encoded by *EjFAD8* was cloned by homologous recombination into the overexpression vector pCAMBIA2300 with the CaMV 35S promoter (restriction digest sites BamH1 and Xba1), and the resulting plasmid was then transferred into the *Agrobacterium tumefaciens* strain (GV3101). WTs were transformed using the floral-dip method to obtain T0 generation transgenic lines [47]. Overexpressing T1 generation seeds were obtained by sowing on 1/2 MS solid medium containing 50 mg/L kanamycin. The transgenic lines used in the experiment were T3 generation seeds.

### 4.4. Subcellular Localization of EjFAD8

To understand the subcellular localization of EjFAD8, we cloned the coding sequence of EjFAD8 (without termination codon) into the pCAMBIA2300--35S::EGFP vector, and the restriction digest sites were BamH1 and Xba1. The EjFAD8-EGFP fusion protein and the empty vector were then transformed into the *Agrobacterium tumefaciens* strain (GV3101) and injected into tobacco (*Nicotiana benthamiana*) leaves using a syringe [48]. The EGFP signal in tobacco leaves was later observed under a confocal laser scanning microscope (Leica SP8, Wetzlar, Germany).

### 4.5. Total RNA Extraction and Quantitative Real-Time PCR Analysis

Total RNA was extracted from loquat leaves and *Arabidopsis* leaves using an RNA extraction kit (Tiangen, Beijing, China) according to the instructions for use. First-strand cDNA was synthesized from total RNA using M-MLV (RNase H-) reverse transcriptase (TaKaRa, Beijing, China). qPCR was performed using qTOWER^3^ G (Analytik Jena, Jena, Germany) to measure changes in gene expression in loquat and in *Arabidopsis thaliana*. The qRT-PCR analysis was performed using Novostar-SYBR Supermix (Novoprotein, Shanghai, China). The condition for the qRT-PCR was as follows: initial denaturation for 5 min at 95 °C, followed by 41 cycles of 95 °C for 15 s, 56 °C for 30 s, and 72 °C for 30 s. All primer sequences are listed in Appendix A. The loquat *EjActin* gene and *Arabidopsis AtActin* gene were used as internal controls. Transcript levels are expressed as relative values, with a control expression value of 1.

### 4.6. Low-Temperature Stress Treatments

Seeds of WT and transgenic lines were sown in square 1/2 MS medium. The plants were incubated vertically at a low temperature of 4 °C for 10 days to observe the effect of low temperature on plant growth. WT and transgenic lines were sown in substrate for 4–5 weeks and were the transferred to treatment at 0 °C. The desired samples were extracted at different times to determine various indicators of low-temperature stress or tolerance. *Arabidopsis* were used as controls under normal growth conditions (23 °C). To determine the electrolyte leakage from *Arabidopsis* leaves at low temperatures, 0.2 g of leaves were placed in 20 mL of distilled water overnight, and then their conductivity (E1) was measured using a conductivity meter (DDS-309^+^, Chengdu, China). The leaves were then boiled for half an hour and the conductivity (E2) was recorded after the solution had cooled to room temperature. The relative electrolyte leakage was (E1/E2) × 100%.

### 4.7. Measurement of the Content of ROS and Chlorophyll

The accumulation of H_2_O_2_ and O^2−^ in *Arabidopsis* leaves was determined by the nitroblue tetrazolium (NBT, 0.5 mg/mL) and 3,3-diaminobenzidine (DAB, 1 mg/mL) uptake methods previously described [49,50], respectively. Fresh leaf samples (0.2 g) with veins removed were placed in 10 mL of 95% ethanol and left to stand for 24 h. When the leaves turned white, absorbance of extracts was measured using a spectrophotometer at 665 nm and 649 nm and calculated according to the formula in the literature [51].

### 4.8. RNA-Seq Analysis

Loquat leaves before and after treatment were collected via liquid-nitrogen flash freeze and stored in a −80 °C refrigerator. The stored samples were then sent to Nanjing Personal Gene Technology Co., Ltd. (Nanjing, China) for RNA-seq. After obtaining data from the company’s transcriptome sequencing, the best feature annotations were obtained by comparing its single gene sequences with the Protein Data Bank obtained from NR, SwissProt to identify differentially expressed genes (DEGs). The required genes were further selected by means of the LOG2FC criteria.

### 4.9. Untargeted Relative Quantitative Lipidomics Analysis

Further metabolite extraction and LC-MS/MS analysis were carried out on loquat and *Arabidopsis* leaves before and after treatment. The samples were sent to SHANGHAI BIOTREE BIOTECH CO., Ltd. (Shanghai, China) to conduct lipidomics analysis. A 25 mg sample was placed in a centrifuge tube and 200 μL of water and 480 μL of extraction solution (MTBE:MeOH = 5:1) were added. The grinding process was carried out by placing steel beads for 4 min and sonicating for 5 min in an ice water bath, repeated 3 times. After 1 h at −40 °C, the samples were centrifuged at 4 °C and 3000 rpm for 15 min. An amount of 350 μL of supernatant was placed in a centrifuge tube and dried under vacuum. Then, 200 μL of the solution (DCM:MeOH = 1:1) was added for re-dissolution, vortexed for 30 s, sonicated in an ice water bath for 10 min, and then centrifuged at 4 °C, 3000 rpm for 15 min. Finally, 100 μL of the supernatant was placed in the injection vial and tested on the machine.

We used an UHPLC system (1290, Agilent Technologies, Santa Clara, USA), equipped with a Phenomen Kinetex C18 (2.1 × 100 mm, 1.7 μm) liquid chromatographic column for the chromatographic separation of the target compounds. Liquid chromatography phase A: 40% water and 60% acetonitrile solution containing 10 mmol/L ammonium formate; phase B: 10% acetonitrile, 90% isopropanol solution with 50 mL of aqueous 10 mmol/L ammonium formate solution added per 1000 mL, using gradient elution. 0~1.0 min, 40% B; 1.0~12.0 min, 40~100% B; 12.0~13.5 min, 100% B; 13.5~13.7 min, 100~40% B; 13.7~18.0 min, 40% B. The column temperature was 55 °C. The auto-sampler temperature was 4 °C and the injection volume was 2 μL (pos) or 2 μL (neg), respectively.

The mass spectra were converted from raw to mzXML format using ProteoWizard software. After a series of analyses, lipid identification was carried out based on XCMS software, a self-authored R package, and the lipidblast database. The difference was calculated by fold change and LOG2FC of the quantity of lipids before and after sample treatment.

### 4.10. Statistical Analysis

All experiments consisted of three independent replicates, and data are shown as mean ± SD. Statistical significance, * *p* < 0.05, ** *p* < 0.01 and *** *p* < 0.001, was determined by one-way ANOVA using SPSS 22.0 (SPSS Institute, Inc., Chicago, IL, USA).

## Figures and Tables

**Figure 1 ijms-24-06946-f001:**
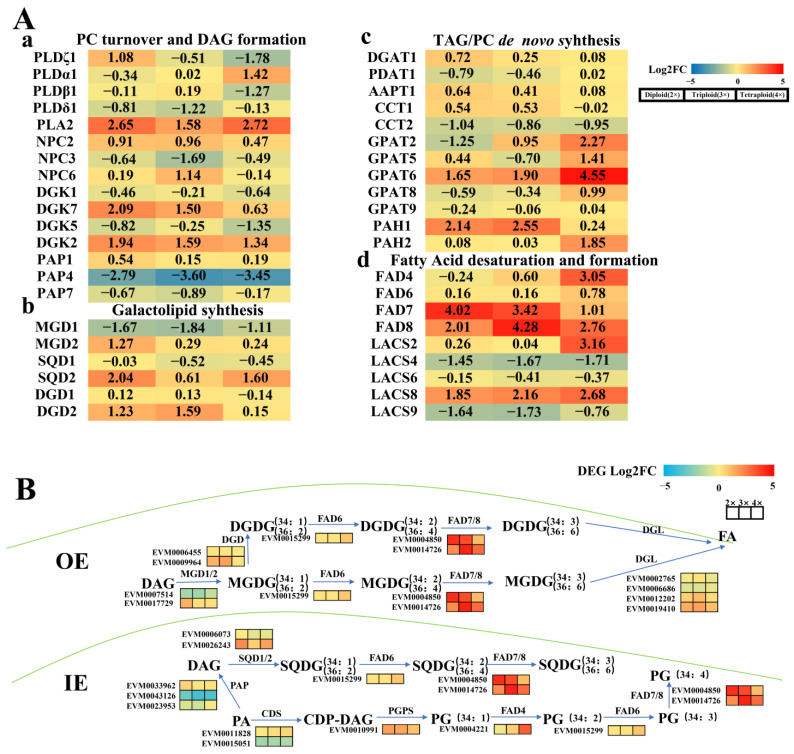
Differentially expressed genes (DEGs) and metabolic networks of major lipid metabolic pathways in loquat leaves under low-temperature stress (−3 °C). (**A**) Differentially expressed genes (DEGs) in loquat leaves under low-temperature stress (−3 °C). The number in each color block represents the Log2 (fold-change) of the corresponding genes, and the negative number represents down-regulated DEGs. The color scale was provided. Red indicates a higher expression level, and blue indicates a lower expression level. (**B**) Gene-metabolite network illustrating membrane lipid metabolism in loquat under low-temperatur stress. The glycerolipid synthesis pathway is mapped, and the genes and lipid metabolites involved are symbolized. The relative expression levels of selected genes are labelled as heat map icons. The color scale is provided. Red indicates a higher expression level, and green indicates a lower expression level. OE, outer envelope; IE, inner envelope; PA, phosphatidic acid; CDP-DAG, cytidine diphosphate diacylglycerol; DAG, diacylglycerol; FA, fatty acids; DGDG, digalactosyl diacylglycerol; MGDG, monogalactosyl diacylglycerol; SQDG, sulfoquinovosyl diacylglycerol; PG, phosphatidylglycerol; PLD, phospholipase D; PLA, phospholipase A; NPC, non-specific phospholipase C; DGK, diacylglycerol kinase; PAP, phosphatidic acid phosphatase; DGAT, diacylglycerol acyltransferase; PDAT, phospholipid diacylglycerol acyltransferase; AAPT, anolamine phosphotransferase; CCT, choline phosphate cytidylyltransferase; GPAT, glycerol-3-phosphate acyltransferase; PAH, phosphatidate phosphatase; MGD, monogalactosyl diacylglycerol synthase; SQD, sulfoquinovosyl diacylglycerol Synthase; DGD, digalactosyl diacylglycerol Synthase; LACS, long chain acyl-CoA synthetase; DGL, diacylglycerol lipase; CDS, cytidylyltransferase.

**Figure 2 ijms-24-06946-f002:**
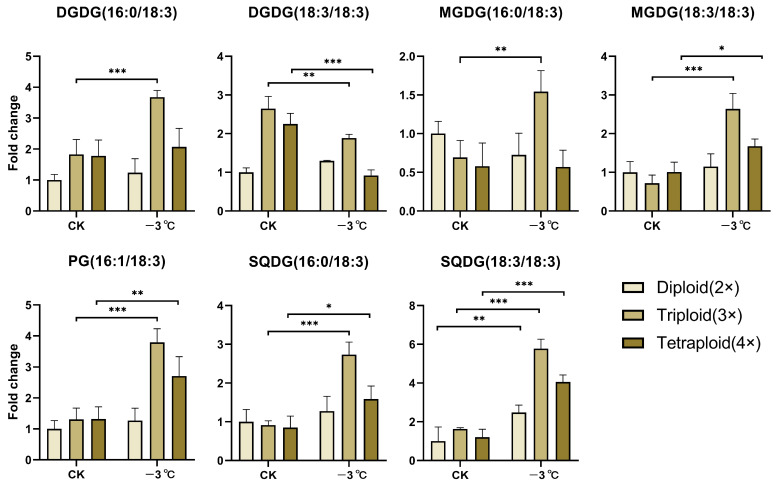
Changes in the quantity of membrane lipids associated with EjFAD8 in different ploidy loquats under low-temperature stress. Error bars, ±SD. Each analysis was repeated with three biological replicates. Asterisks denote significant differences: * *p* < 0.05; ** *p* < 0.01; *** *p* < 0.001.

**Figure 3 ijms-24-06946-f003:**
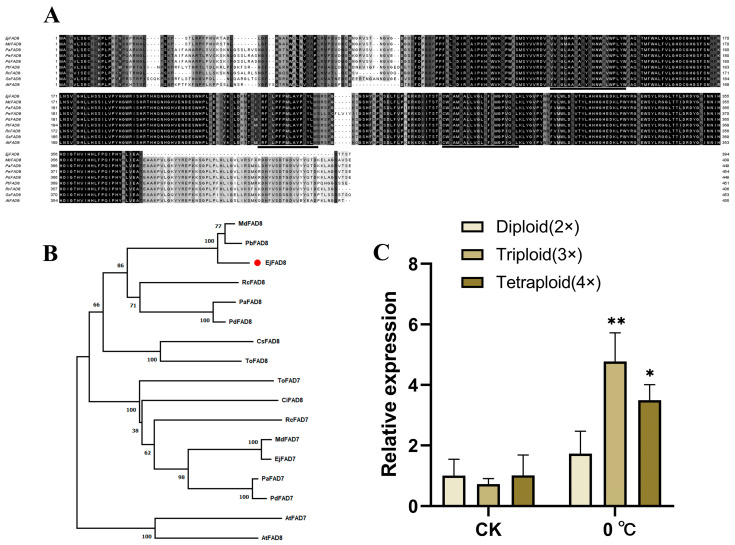
Identification of *EjFAD8*. (**A**) Multiple alignment of amino acid sequences of EjFAD8 with MdFAD8 (*Malus domestica*), PaFAD8 (*Prunus avium*), PmFAD8 (*Prunus mume*), PdFAD8 (*Prunus dulcis*), PtFAD8 (*Populus trichocarpa*), RcFAD8 (*Rosa chinensis*), GsFAD8 (*Glycine soja*), and AtFAD8 (*Arabidopsis thaliana*) proteins. The underlined region is the transmembrane structural domain. (**B**) Phylogenetic analysis among EjFAD8 and FAD7/8 proteins from other plant species. (**C**) Relative expression analysis of *EjFAD8* in different ploidy loquats at 0 °C. Error bars, ±SD. Each analysis was repeated with three biological replicates. Asterisks denote significant differences (as compared with the diploid loquat): * *p* < 0.05; ** *p* < 0.01.

**Figure 4 ijms-24-06946-f004:**
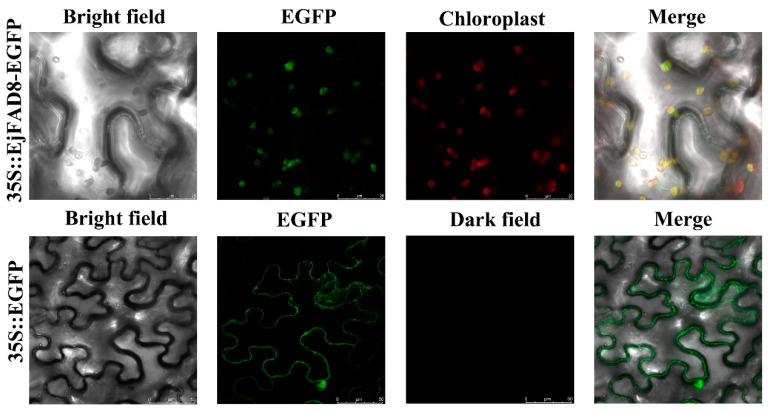
The subcellular localization of EjFAD8 in *Nicotiana benthamiana* epidermal cells. EGFP, enhanced green fluorescent protein.

**Figure 5 ijms-24-06946-f005:**
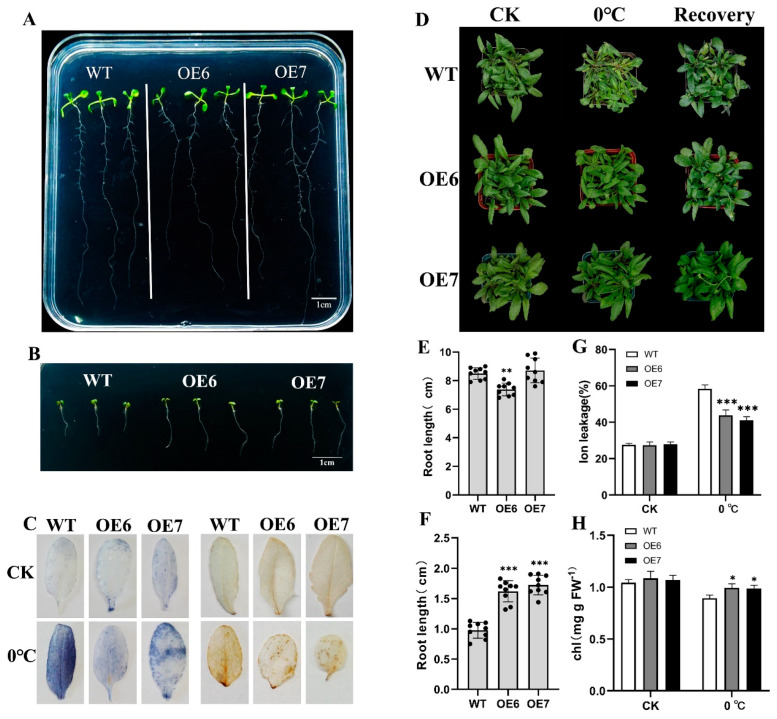
The phenotypes of wild-type and transgenic *Arabidopsis* at low temperature. (**A**) Root growth of WT and transgenic lines at 23 °C for 10 days. (**B**) Root growth of WT and transgenic lines at 4 °C for 10 days. (**C**) NBT and DAB staining. (**D**) WT and transgenic lines were treated at 0 °C for 5 days and then recovered at 23 °C for 3 days. (**E**) Root length of WT and transgenic lines at 23 °C for 10 days. (**F**) Root length of WT and transgenic lines at 4 °C for 10 days. Data are shown as mean ± SD of nine wild-type or T3 transgenic plants (*n* = 9). (**G**) Ion leakage. (**H**) Total chlorophyll content. Error bars, ±SD. Each analysis was repeated with three biological replicates. Asterisks denote significant differences (as compared with the WT): * *p* < 0.05; ** *p* < 0.01; *** *p* < 0.001. WT, wild-type. OE, overexpression lines. CK, Control Check.

**Figure 6 ijms-24-06946-f006:**
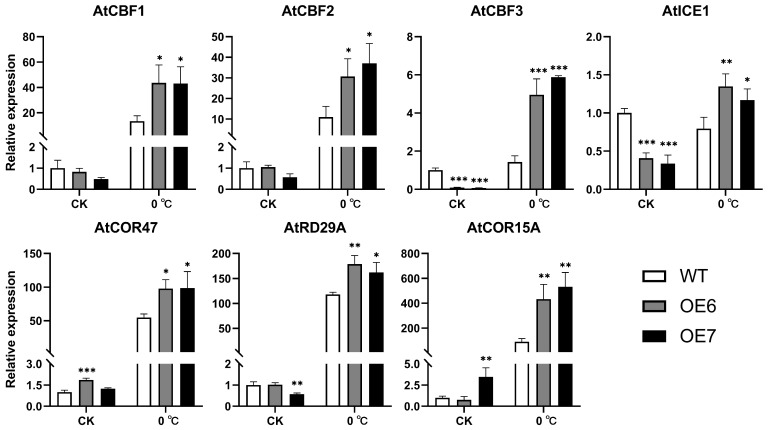
Expression levels of low-temperature stress-related genes in WT and transgenic *Arabidopsis* under low-temperature stress. Error bars, ±SD. Each analysis was repeated with three biological replicates. Asterisks denote significant differences (as compared with the WT): * *p* < 0.05; ** *p* < 0.01; *** *p* < 0.001.

**Figure 7 ijms-24-06946-f007:**
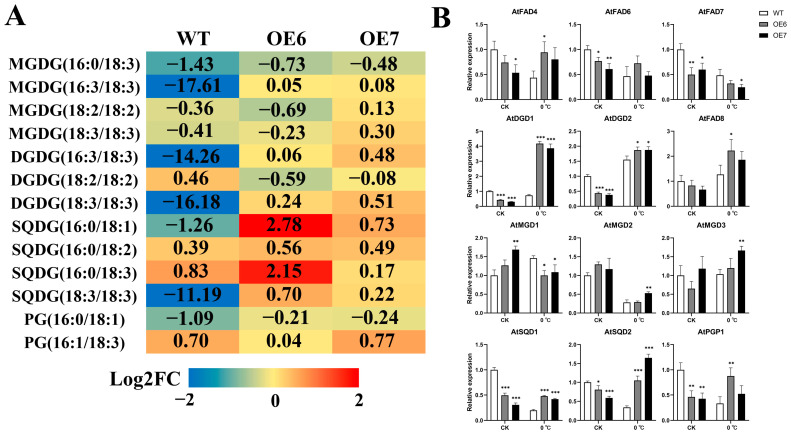
Changes in lipid and lipid-related gene expression at low temperatures in WT and transgenic *Arabidopsis*. (**A**) Changes in *FAD8*-related lipid quantity in *Arabidopsis* leaves under low-temperature stress (0 °C). The number in each color block represents the Log2 (fold-change) of the lipid, with negative numbers representing down-regulated lipids. Red indicates an increase in quantity and blue indicates a decrease in quantity. (**B**) Expression levels of lipid-related genes in WT and transgenic lines under low-temperature stress (0 °C). Error bars, ±SD. Each analysis was repeated with three biological replicates. Asterisks denote significant differences (as compared with the WT): * *p* < 0.05; ** *p* < 0.01; *** *p* < 0.001.

## Data Availability

The dataset presented in this study can be found in the online repository. The names and login numbers of the repositories for transcriptomic data can be found below: https://www.scidb.cn/anonymous/N1pyeUly (accessed on 12 January 2022). The names and login number of the repositories for lipidomic data can be found below: https://www.scidb.cn/anonymous/cTZqNmpt (accessed on 16 February 2022).

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
