# Peer review of "EjFAD8 Enhances the Low-Temperature Tolerance of Loquat by Desaturation of Sulfoquinovosyl Diacylglycerol (SQDG)"

_ijms, 2023, doi:10.3390/ijms24086946_

Round 1

Reviewer 1 Report

Opinion about the Manuscript ID: ijms-2265615

Title: «EjFAD8 enhances the low temperatures tolerance of loquat by desaturation of sulfoquinovosyldiacylglycerol (SQDG)».

Authors: Xun Xu, Hao Yang, Xiaodong Suo, Mingxiu Liu, Danlong Jing, Yin Zhang, Jiangbo Dang, Di Wu, Qiao He, Yan Xia, Shuming Wang, Guolu Liang and Qigao Guo*

The article presents very interesting data on the role of FAD8 desaturase for the resistance of Eriobotrya japonica plants to low temperatures and provides undoubted experimental evidence. The work is of undoubted interest and worthy of publication.

However, it is worth paying the attention of the authors to linguistic mistakes and some technical issues.

I bring something in this text. The rest is noted in the manuscript of the article.

________________

Sulfoquinovosyl diacylglycerol should be written in two words.

Lines 30-31

"On the other hand, desaturation of fatty acids increased the content of SQDG to maintain the stability of photosynthesis under low temperatures."

It is not clear how desaturation of fatty acids leads to an increase in SQDG?

Lines 62-64

"The ω-3 fatty acid desaturases, representative of the fatty acid desaturase family, synthesize diene fatty acids (DA) into triene fatty acids (TA), in which FAD3 is located in the endo plasmic reticulum (ER) and FAD7 and FAD8 in the plastid [16–18]".

Most likely they do not synthesize, but transform ...

Lines 84-86

"The induced CBF protein binds to the cis-element present in the COR gene promoter and activates the expression of the COR gene, thus increasing the cold tolerance of the plant [28]".

It should be noted that cold acclimation is a process used by most temperate plants to cope with freezing stress. In this process, the expression of cold-responsive (COR) genes is activated and the genes undergo physiological changes in response to the exposure to low, non-freezing temperatures and other environmental signals. We should talk about COR genes, not just one gene.

Line 86

"… Chloroplast, as one of the cryoreceptors in plants…"

Looks a little strange.

Lines 89-90

"Fatty acid desaturation, as an important REDOX reaction plays an important role in chloroplast regulation".

What is meant by chloroplast regulation?

Lines 96-97

"We then conducted a series of studies on its physiology, biochemistry and molecular mechanisms".

Molecular mechanisms of what?

Other comments in the text of the manuscript.

______________________

The work requires editing the English language and minor technical corrections. After that, in my opinion, the work can be accepted for publication.

Reviewer 2 Report

Dear Editor, Dear authors,

The manuscript characterizes protein FAD8 and its coding gene from loquat in respect to the sensitivity to low temperatures. The characterization is performed in physiological, lipidical, cytological and genetical levels using three plants – loquat, tobacco, and Arabidopsis thaliana.

The protein function as desaturase activity in low temperatures is already known, so the work is not so much scientifically relevant, but the characterization of protein in loquat was performed for the first time.

The work is quite comprehensive, so only some minor changes should be considered in my opinion. The same review as here I also attached as MSWord file. 

Abstract.

Line 24. Should be SQDG (16:0/18:1; 16:0/18:3), in my opinion, instead of SQDG (16:0/18:3; 18:3/18:3).

Line 26. “ICE-CBF-COR pathway” should be, in my opinion, instead of “CBF-COR related pathway”.

Line 29. In my opinion, there are no direct evidence provided in the manuscript, that namely “desaturation of fatty acids increased the expression of CBF-COR (again ICE-CBF-COR, probably, should be here) genes”. These events may occur in parallel and ICE-CBF-COR pathway gene expression could be enhanced also through the other mechanisms.

Line 32. EjFAD8 gene was not overexpressed in loquat in this work.

Introduction

The first paragraph of the Introduction, in my opinion, should be reconsidered. The references 1,2 do not describe the long history of loquat cultivation in China embracing 2200 years (the number not mentioned in the references 1,2?). Moreover, the reference 2 claims, that loquat originated in south-central China, not south-western China. The references 5,6 are not very proper, in my opinion, because they are more devoted to the post-harvest fruit storage investigations, but less to the flower and juvenile fruit development susceptibility to low temperature. In the reference 7 cold is not considering.

The sentence “When plants are exposed to low temperatures, the cell membrane first receives the cold signal and transmits it to the nucleus, which triggers the above-mentioned changes in the expression of cold-responsive genes” (lines 48-50) have two stylistics drawbacks: membranes do not transmit signals to nucleus themselves and changes in the expression of cold-responsive genes were not mentioned above at least in the Introduction.

Another paragraph, which should be a little reorganized, in my opinion, that is describing FAD8 functions (lines 65-75). I am doubt about the conversion of palmitoleic acid by FAD8, because palmitoleic acid is omega 7 acid and FAD8 is omega 3 acid desaturase. At least there is no evidence of such activity in the presented references [19,20]. As well as other mentioned fatty acid, linoleic acid is omega 6 acid, and again this should be poor substrate for FAD8, in my opinion. Also, the evidence for linoleic acid as substrate for FAD8 are scarce in the references 19, 20. Another comments are the lack of reference to the statement “For example, AtFAD8 was up-regulated in Arabidopsis at 8°C and down-regulated at 30°C” (lines 71-72) and it is not absolute true to say that ZmFAD8 is down regulated at 5 oC, because the decrease presented in the reference 23 was not statistically significant (but up-regulation at 50 oC, yes, it was significant).

Results

As the work was performed with three plant species (loquat, tobacco, and Arabidopsis) I suggest in all figure titles indicate plant species to which the figure information is relevant.

Lines 116-118. How the significance of differences between ploidy in RNA DEGs were tested (Fig. 1A) to prove the statement, that “EjFAD8 was more significantly up-regulated in the triploid loquat than in its parental loquat”?

Lines 129-130. MGDG (16:0/18:3) is omitted from the list. Is it because of less significance (p<0.01 vs p<0.001)?

Lines 134-135. The proposition is not absolutely correct, because, as you claimed in lines 114-115 also EjFAD7 was up-regulated in the response to low temperature. So, probably, both proteins were responsible for an increase in unsaturated fatty acids of MGDG, DGDG, PG, and SQDG in loquat.

Fig.1 caption. Line 137. The first two sentences in the caption of Figure 1 are not necessary.

Line 147. The abbreviations of ferments are not described.

Fig.2 caption. Line 149. The phrase “of different ploidy” doubles.

Lines 153-160. In my opinion more attention should be paid to the molecular characterization of your investigating product. First, it would be better to present the sequence of your 1155 bp product or the link to the sequence. After that, it is needed more evidence for the name provided as FAD8. What evidence links your obtained product more with FAD8, compared to other similar proteins, particularly FAD7, considering, that both proteins are very similar? Also, proteins used for the comparison in multiple alignment and phylogenetic analysis (Fig. 3 A and B) must be identified somehow more. If they are from the NCBI database, ID numbers should be provided. On what basis all proteins for alignments were assigned as FAD7 or FAD8?

Fig.3 caption. Line 168. The first two sentences in the caption of Figure 3 are not necessary.

Lines 175-181. Subcellular localization of EjFAD8. How did you decide, that 35S::EjFAD8-EGFP localizes in the chloroplasts? Did you use some markers of chloroplasts?

Line 195. “lines s” should be “lines”.

Line 216. “The Phenotypes” should be “The phenotypes”. Also, I had better prefer to expand the title of caption including the plant name. Almost nowhere no explanation what means CK?

Line 232. We could not confidently propose, that AtICE1 transcription level was lower in the control variant as from the Fig. 6 seems, that the difference was not statistically significant.

Line 234-235. Though the expression level of AtKIN1 was higher, but not statistically significant (Fig. 6). You must consider this, writing about significance on line 235.

The statement in lines 249-250 should be reconsidered, because MGDG (16:0/18:3) was reduced in OE6 and OE7. Also, MGDG (18:3, 18:3) was reduced in OE6. Regarding MGDG (16:3, 18:3), log2FC 0,05 and 0,08 hardly can be accepted as increase, due to small values.

Line 254. „AtAFD8“ should be „AtFAD8“.

Lines 256-258. From Fig. 7 B we can see down regulation of gene expression and for other desaturases (AtFAD4, AtFAD6), other ferments such as AtMGD2, AtSQD1 and 2, and AtPGP1, and even maybe other unanalyzed desaturases such as AtFAD3 could be downregulated, so it is incorrect, in my opinion, to claim, that only AtFAD7 lead to a reduction in the number of lipids in WT.

Lines 265-268. The statement is doubtful, because only AtFAD4 and only in one overexpression line (OE6) was significantly up regulated, compared to control at 0 oC.

Discussion

For what reason you did not consider another important desaturase FAD3 in your work, though mentioned it in the introduction (Line 63)?

Lines 287-288. The reference 33 do not consider photosynthetic efficiency, so it is not appropriate here.

Lines 325-326. Does indeed FAD8 gene is localized in chloroplast?

Line 338. The reference 39 does not describe FAD8 protein. Improper citation.

Line 388. I think, that there is a lack of evidence, that FAD8 directly participates in the pathway ICE-CBF-COR, so may be better to say affecting“, „inducing“ instead of „participating“.

Materials and methods

Would be nice to know, how old loquat scions were at the time of temperature treatment, i.e. how much time passed from the grafting?

Line 401. Suppose here Arabidopsis thaliana should be rather than Nicotiana benthamiana.

Lines 411-413. The primary selection of EjFAD8 is described not very clear, at least for me. Because there is no number EVM0014726 in NCBI database, I suppose that this number comes from the sequencing results. Also, it would be good to clarify what means loquat database mentioned in line 413. What criteria/programs did you use assigning sequences to FAD8?

Line 421. Did you use BLAST for the search of other similar sequences?

Lines 446-447. Could you, please, provide the sources of the primer sequences. If it generated by you, could you, please, specify the program used for generating primer sequences.

Line 467. “Absorbance of extracts” sounds better.

How many biological replicates were sent for sequencing?

Good luck,

Sincerely Yours,

Reviewer
